# New approaches on composite designs for Response Surface Methodology

**Despina E. Athanasaki, Stelios D. Georgiou** *, **Stella Stylianou**

School of Science, RMIT University, Melbourne, VIC, Australia

* stelios.georgiou@rmit.edu.au

## Abstract

Recent attention has increasingly focused on the significance of Definitive Screening Designs (DSDs), originally introduced by Jones and Nachtsheim (2011), as a compelling alternative to traditional designs bib Response Surface Methodology (RSM). This paper introduces two novel composite techniques aimed at enhancing design efficiency and elevating D-values. By utilizing orthogonal matrices and integrating axial components from either simple orthogonal designs or the block orthogonal designs detailed in the work of Alrweili et al. (2020), new design matrices are constructed based on established composite design principles. Notably, the novel designs presented in this manuscript surpass previously documented designs in the existing literature in terms of design efficiency and robustness.

**Data Availability Statement:** All relevant data are in the manuscript.

**Funding:** The author(s) received no specific funding for this work.;

## 1 Introduction

The main objective of Response Surface Methodology (RSM) is to streamline and enhance the optimization of processes and systems. This is achieved through the construction of sophisticated models that capture the intricate interplay between input factors and output responses. Experimental designs within Response Surface Methodology (RSM) are intricately designed frameworks aimed at systematically exploring and understanding the relationships between input variables and corresponding output responses. These designs are meticulously crafted to efficiently collect data points, allowing for the accurate construction of response surfaces and the derivation of predictive models. By strategically selecting factor levels and configurations, experimental designs in RSM enable researchers to extract maximum information from a minimal number of experiments, facilitating the process of optimization and decision-making.

The response surface methodology process can be outlined as follows. Begin by identifying the critical factors that influence a response variable. Determine the specific levels at which these factors need to be investigated. Select an appropriate experimental design and conduct the experiments according to the plan. Thoroughly document the data and responses obtained from the experiments. Develop a response model by employing regression analysis on the collected data. Assess the significance and fitness of the model. Utilize the established model to identify the factor levels that lead to the desired outcomes. Verify these conclusions through additional supplementary experiments to ensure their validity. Evaluate the impact of variations on the outcomes and apply the optimized settings to improve the process. Maintain continuous vigilance over the ongoing performance to sustain the desired results.

**Competing interests:** The authors have declared that no competing interests exist.

Several key questions must be addressed to optimize the necessary experimentation process. These questions may include the following: What design is best suited for specific factors and research hypotheses? Particularly, which design is effective in accommodating a second-order model while minimizing the necessary number of experimental runs? Throughout time, statisticians and computer scientists have introduced numerous designs. An ambitious goal is to create a full second-order model without employing a full factorial design. In Response Surface Methodology (RSM), practitioners frequently face the challenge of balancing the desire for higher D-values (as outlined in Eq (3)) to indicate informative and robust designs with the need for economical run sizes and other constraints present in actual experimental conditions. Strategically optimizing the D-value, along with other factors such as the number of experimental runs, enables the development of a practical experimental design that provides valuable insights and improves the investigated process or system.

One of the classical designs found in the literature is the Central Composite Design (CCD), which was originally proposed by Box [1]. The CCD serves as a prime example of a composite design, encompassing three sets of runs identified as *F*, *A*, and *C*, and laid out in the subsequent matrix arrangement:

$$X = \begin{pmatrix} F \\ A \\ -A \\ C \end{pmatrix}.$$ 

(1)

In this configuration, the component *F* represents the factorial part of the design, incorporating two levels for each factor involved in the experiment. These levels are commonly denoted as High/Low, $+/-$, 0/1, or $+1/-1$. The number of runs in the factorial part is denoted by $n_f$. The axial component of the design, *A*, adopts the format of a diagonal matrix where an unspecified value *a* is placed along the diagonal, while the remaining matrix elements are set to 0. The experimenter is responsible for determining the optimal value of *a*, with the goal of maximizing the potential efficiency of this specific segment within the design. Then number of runs in the axial part is denoted by $n_a$. Lastly, the *C* part encompasses the center points of the design. Replicating a central point multiple times contributes to enhancing experimental precision, and these repetitions constitute the *C* part. The number of runs in the centre points part is denoted by $n_c$. The total number of non-centre points in the design is given by $n_r = n_f + n_a$. Enhancements and variations of these designs have been detailed in numerous scholarly articles. For instance, Georgiou et al. [2] introduced modifications aimed at boosting the efficiency of the resulting designs. In their initial study, the authors made adjustments to the axial points by substituting them with points situated within a hypercube that encompasses a sphere with a radius equivalent to the parameter *a*. This innovative approach allowed them to decrease the required number of experimental runs, while simultaneously elevating the D-value of the design concerning the complete second-order model. For more recent research in this direction, interested readers are encouraged to explore the work by Liu et al. [3], Ares and Goos [4], Jones et. al. [5], and the references therein. In this research paper, the authors not only delve into the augmentation of the generated designs but also delve into the concept of projectivity, establishing noteworthy findings regarding the efficiency of the designs.

Considerable attention has been directed toward the widely acclaimed Definitive Screening Designs (DSD) developed by Jones and Nachtsheim [6]. DSDs call for an analytical approach that differs from the norm applied to other designs. The term "screening" in their name indicates their purpose of identifying variables with significant linear impacts. DSDs are

constructed with three levels for all the continuous factors in the design and can be used to obtain a cost-effective approximation of the actual response surface. The rationale behind screening lies in the recognition that fitting an entire quadratic polynomial model would demand an excessive number of runs due to the presence of numerous intriguing factors. Instead, a model encompassing linear terms, some two-factor interactions, and possibly quadratic terms (a second-order model) is employed, as elaborated by Jones and Nachtsheim [7]. A distinguishing attribute of this design is its utilization of $2k + 1$ runs, with $k$ representing the number of factors. Consequently, $k$ signifies both the count of main effects and the count of quadratic effects. By employing a Definitive Screening Design (DSD), it becomes possible to scrutinize the effects of numerous variables through a compact experiment. In essence, DSDs represent an advancement over conventional screening designs, such as the Plackett-Burman design, allowing for the detection of non-linear responses and the avoidance of factor confounding, as noted by Yu et al. [8].

Screening designs are aimed at pinpointing the significant main effects from a vast number of potentially influential effects while minimizing costs. These designs traditionally employ two levels and the minimum number of runs. DSD takes this objective even further by enabling the estimation of quadratic effects and even the complete second-order model without requiring additional experimental runs, albeit in cases where the screening process identifies only three active main effects. Liu et al. [3] propose a novel composite design strategy. They suggest replacing the factorial part of the Central Composite Design (CCD) with DSD. They extensively discuss the design, particularly focusing on the axial part that employs classical diagonal matrices $A$ and $-A$, as previously discussed. This approach allows for the addition of runs, often leading to a significant reduction in the average prediction variance. Their design construction takes the form of Eq (1) with matrix $F$ being the desirable $DSD$.

In scenarios involving an odd number of factors, Alrweili et al. [9] presented the idea of a modified axial part structure that comprises blocks of pre-existing orthogonal designs. This necessitates the addition of an extra column and rows to maintain orthogonality. The proposed designs in their work incorporated a composite design matrix $X$ as in Eq (1) but with the part $A$ in matrix $X$ being an axial part of a different structure. Some fundamental criteria that it is desirable to be maintained in all the constructed designs for response surface methodology include: a) Orthogonality of the main effects (and other effects in the model, if feasible), b) Minimum total number of experimental runs, and c) High D-value (see Eq (3)) for the complete second-order model.

## 2 Evaluation criteria

This study aims to explore innovative approaches for developing designs for a second-order Response Surface Methodology (RSM). The proposed methodologies blend established techniques, enhancing both the factorial and axial components of composite designs. When examining the behavior of the response variable, in the presence of k potentially influential factors, the conventional RSM approach initiates with an experimenter conducting a screening design and fitting a first-order (linear) model to the obtained data. Subsequent experiments are then required to fit a more suitable model. As we have discussed in the previous session, several research papers have introduced and employed new designs for RSM, second-order models, and even higher-order polynomial designs. In this context, we will employ the following second-order model:

$$\mathbf{y} = \beta_0 + \sum_{i=1}^{k} \beta_i \mathbf{x_i} + \sum_{i=1}^{k} \beta_{ii} \mathbf{x_i}^2 + \sum_{i=1}^{k-1} \sum_{j=i+1}^{k} \beta_{ij} \mathbf{x_i} \mathbf{x_j} + \boldsymbol{\varepsilon}, \tag{2}$$

where **y** signifies the response vector, while $\mathbf{x_i}$ represents the column within the model matrix that pertain to the main effect of factor $i$. Additionally, $\mathbf{x_i}^2$ designates the column in the model matrix specifically associated with the pure-quadratic effect of variable $i$, while $\mathbf{x_i}\,\mathbf{x_j}$ denotes the column in the model matrix that correspond to the two-factor interaction effect of variables $i$ and $j$. $\beta_0$ represents the unknown constant intercept, $\beta_i$ corresponds to coefficient of the main effect $\mathbf{x_i}$, $\beta_{ii}$ characterizes the quadratic effects, and $\beta_{ij}$ signifies the coefficient of the two-factor interaction effect of $\mathbf{x_i}\mathbf{x_j}$. The presence of the error vector $\boldsymbol{\varepsilon}$ accounts for variability in the model.

D-optimal design is a versatile and efficient approach to optimal experimental design, aiming to maximize the determinant of the information matrix associated with a particular model. This method excels in selecting the most informative experimental runs from a given set, accommodating any combination of factors, factor levels, and model complexities. Despite its flexibility and advantages, D-optimal design has certain limitations. It necessitates an accurate initial model estimation and a reasonable understanding of the factor ranges, making it sensitive to inadequate model assumptions and outliers in the data. Therefore, while powerful, D-optimal design requires careful consideration and data preparation to harness its full potential in experimental design and optimization.

In this paper, the D-efficiency of the information matrix of the full second-order model matrix is used to evaluate the design. A function of the determinant of the information matrix is often used for this evaluation. The D-value criterion that will be implemented here is as described by Nguyen and Lin [10]. We denote the information matrix $X_{inf} = X'_p X_p$, where $X_p$ denotes the model matrix, $n$ is the run size in the design (and model) matrix, and $p$ denotes the model's parameters. Note that $p = 1 + 2k + k(k-1)/2$ in the case of the full second-order model. We are looking for the greatest possible $D - values$ using the following variation from the literature:

$$D - value = \frac{\left(10^3 \left| X'_p X_p \right|\right)^{\frac{1}{p}}}{n} \tag{3}$$

We chose to use the D-efficiency form mentioned in Eq (3) so that readers can easily compare our results with those in existing literature. This helps create a common basis for understanding and discussing our findings in relation to other studies. Our goal is to contribute to the ongoing scientific conversation and make it simpler for everyone to interpret and compare results.

## 3 Design components

In this section, we provide a summary of the essential components of the composite designs that are required for implementing the suggested methodologies in constructing the new designs we proposed.

### 3.1 Orthogonal designs (*ODs*)

An *orthogonal design* of order $n$ is denoted $OD(n; a_1, a_2, \ldots, a_m)$, having the variables ($\pm x_1$, $\pm x_2, \ldots, \pm x_m$), is a square matrix of order $n$ with entries $\pm x_k$. An *orthogonal design* has its rows and columns pairwise orthogonal. This means that any two columns (or rows) have their dot product equal to zero and thus the design matrix $D = OD(n; a_1, a_2, \ldots, a_m)$ satisfies

$$DD' = D'D = (a_1 x_1^2 + a_2 x_2^2 \cdots + a_i x_i^2) \times I_n,$$

where $\pm x_1, \pm x_2 \ldots, \pm x_m$ are commuting variables, and $I_n$ is the identity matrix of order $n$.

An *orthogonal design* $D = OD(n; a_1, a_2, \ldots, a_m)$, can have at most $m = \rho(n)$ variables, where

$$\rho(n) = 8c + 2^d, \quad n = 2^t b, \ b \ odd, \quad t = 4c + d, \quad 0 \le d < 4. \tag{4}$$

For more in-depth information on orthogonal designs, interested readers are encouraged to consult Seberry [11].

**Example 1** Presented below are orthogonal designs of orders 2, 4, 6, and 8, which will be employed in crafting the required axial components within the methodologies outlined in this paper:

$$OD_8(8; 1, 1, 1, 1, 1, 1, 1, 1)$$

$$OD_2(2; 1, 1)$$

$$
\begin{pmatrix}
x_1 & x_2 \\
-x_2 & x_1
\end{pmatrix}
$$

$$
\begin{pmatrix}
x_1 & x_2 & x_4 & x_3 & x_6 & x_5 & x_8 & x_7 \\
-x_2 & x_1 & x_3 & -x_4 & x_5 & -x_6 & x_7 & -x_8 \\
-x_4 & -x_3 & x_1 & x_2 & -x_8 & x_7 & x_6 & -x_5 \\
-x_3 & x_4 & -x_2 & x_1 & x_7 & x_8 & -x_5 & -x_6 \\
-x_6 & -x_5 & x_8 & -x_7 & x_1 & x_2 & -x_4 & x_3 \\
-x_5 & x_6 & -x_7 & -x_8 & -x_2 & x_1 & x_3 & x_4 \\
-x_8 & -x_7 & -x_6 & x_5 & x_4 & -x_3 & x_1 & x_2 \\
-x_7 & x_8 & x_5 & x_6 & -x_3 & -x_4 & -x_2 & x_1
\end{pmatrix}
$$

$$OD_6(6; 4, 1)$$

$$
\begin{pmatrix}
x_1 & -x_2 & -x_1 & x_1 & 0 & x_1 \\
-x_1 & x_1 & -x_2 & x_1 & x_1 & 0 \\
-x_2 & -x_1 & x_1 & 0 & x_1 & x_1 \\
-x_1 & -x_1 & 0 & x_1 & -x_1 & -x_2 \\
0 & -x_1 & -x_1 & -x_2 & x_1 & -x_1 \\
-x_1 & 0 & -x_1 & -x_1 & -x_2 & x_1
\end{pmatrix}
$$

$$OD_4(4; 1, 1, 1, 1)$$

$$
\begin{pmatrix}
x_1 & x_2 & -x_3 & x_4 \\
-x_2 & x_1 & -x_4 & -x_3 \\
x_3 & x_4 & x_1 & -x_2 \\
-x_4 & x_3 & x_2 & x_1
\end{pmatrix}
$$

## 3.2 Weighing matrices

A *weighting matrix*, denoted as $W = W_{(n, w)}$, is a square matrix of order $n$, with entries exclusively from the set $-1, 0, +1$. The rows (and columns) are pairwise orthogonal, thus it satisfies $W'W = WW' = wI_n$, where $w$ signifies the count of non-zero elements present in each row (or column) of the matrix, and $I_n$ denotes the identity matrix of order $n$. The parameter $w$ is referred to as the weight of $W$. Conceptually, one can view $W$ as an orthogonal design where all the variables have been substituted with either ±1 or zero values, facilitating various applications in experimental design and statistical analysis, Koukouvinos and Seberry [12].

**Example 2** An example of a *weighing matrix* of order 4 and weight 3 is

$$W_{(4,3)} = \begin{pmatrix} 1 & 1 & 1 & 0 \\ 1 & -1 & 0 & 1 \\ 1 & 0 & -1 & -1 \\ 0 & 1 & -1 & 1 \end{pmatrix}.$$

Specifically, *circulant weighing matrices* are of high significance due to the simplicity provided by their structure. A square matrix is classified as a *circulant matrix* when each row vector is made up by rotating the previous row vector one element to the right. An example of a *circulant Weighing matrix* of order 6 and weight 4 is

$$CW_{(6,4)} = \begin{pmatrix} -1 & 1 & 0 & 1 & 1 & 0 \\ 0 & -1 & 1 & 0 & 1 & 1 \\ 1 & 0 & -1 & 1 & 0 & 1 \\ 1 & 1 & 0 & -1 & 1 & 0 \\ 0 & 1 & 1 & 0 & -1 & 1 \\ 1 & 0 & 1 & 1 & 0 & -1 \end{pmatrix}.$$

In circulant weighing matrices, the whole matrix can be defined by providing the first row vector called as generator vector. Similarly, a circulant weighing matrix $W(7;4)$ of order 7 and weight 4 can be constructed by the generator vector $(-1, 1, 1, 0, 1, 0, 0)$.

Two special cases, of a *weighting matrix*, that are of particular interest are the *Hadamard matrices* and the *Conference matrices*.

## 3.3 Hadamard matrices

A weighing matrix $H_n$ is called a *Hadamard matrix* if $H_n = W(n, n)$. It is easy to show that a Hadamard matrix $H_n$ is a square matrix of order $n$, it has all entries $\pm 1$ (is a 2-level matrix), and any two of its rows (or columns) are orthogonal to each other (i.e. their dot product equals to zero).

Hadamard matrices have several important properties, which make them useful in our application. Their determinant is equal to the square root of its size. According to Brenner and Cummings [13], a *Hadamard matrix* of order $n$ is a solution to the Hadamard's maximum determinant issue because it gives the maximum determinant among any $n$ complex matrices with components $|a_{i,j}| \le 1$. Obviously, $H_n H_n' = H_n' H_n = nI_n$. There are several methods for constructing Hadamard matrices, including Sylvester's construction, Paley construction, and the Walsh–Hadamard transform (see Hennacy [14]).

**Example 3** A $2 \times 2$ *Hadamard matrix is*:

$$H_{2 \times 2} = \begin{pmatrix} 1 & 1 \\ 1 & -1 \end{pmatrix}$$

Using Sylvester's construction method a $4 \times 4$ Hadamard could be:

$$H_{4 \times 4} = \begin{pmatrix} H_{2 \times 2} & H_{2 \times 2} \\ H_{2 \times 2} & -H_{2 \times 2} \end{pmatrix} = \begin{pmatrix} 1 & 1 & 1 & 1 \\ 1 & -1 & 1 & -1 \\ 1 & 1 & -1 & -1 \\ 1 & -1 & -1 & 1 \end{pmatrix}$$

Similarly, the $8 \times 8$ Hadamard matrix can be constructed using a $4 \times 4$ Hadamard, and so forth.

$$H_{8 \times 8} = \begin{pmatrix} H_{4 \times 4} & H_{4 \times 4} \\ H_{4 \times 4} & -H_{4 \times 4} \end{pmatrix}$$

### 3.4 Conference matrices

Conference matrices $C_n$ can be considered as a special case of the weighing matrices. These matrices can be defined as $C_n = W(n, n-1)$. Conference matrices are a particularly important class of 3-level matrices (Xiao et al. [15]). From the definition of weighing matrices, we can conclude that a conference matrix would have only one zero element per row (or column) while the rest are $\pm 1$. The dot product of any two distinct columns (or rows) of $C_m$ is 0 thus the matrix satisfies $C_n' C_n = C_n C_n' = (n-1)I_n$. For more details we refer the reader to Goethals and Seidel [16].

**Example 4** An example of a conference matrix of order 6 is:

$$C_6 = \begin{pmatrix} 0 & 1 & -1 & -1 & -1 & -1 \\ 1 & 0 & -1 & 1 & 1 & -1 \\ -1 & -1 & 0 & 1 & -1 & -1 \\ -1 & 1 & 1 & 0 & 1 & -1 \\ 1 & -1 & 1 & -1 & 0 & -1 \\ 1 & 1 & 1 & 1 & -1 & 0 \end{pmatrix}.$$

### 3.5 Definitive screening designs

Definitive Screening Designs (DSDs), as introduced by Jones and Nachtsheim [6], is a type of experimental design used for studying quantitative factors. Definitive Screening Designs have many advantages compared to Standard Screening Designs as these were introduced by Plackett and Burman [17].

DSDs are designed in such a way that they exhibit mean orthogonality. This means the design allows for precise estimation of main effects and interactions while minimizing bias due to confounding factors. DSDs have a specific number of runs $n$, which is given by the formula $n = 2m + 1$, where $m$ is the number of quantitative factors being studied. This design is saturated, meaning it allows for the estimation of the intercept, main effects, and quadratic effects without redundancy. Unlike some other experimental designs of Resolution III, DSDs ensure that all main effects are orthogonal to all two-factor interactions. This orthogonality property simplifies the analysis and interpretation of experimental results; see, for example, Anderson and Whitcomb [18]. In DSDs, two-factor interactions are designed in such a way that they are not fully aliased with each other. This means that the interactions are designed to be distinguishable from each other, making it possible to identify some of the significant

interaction effects, see for example Vanaja and Shobha Rani [19]. Quadratic effects are independent of main effects and are distinguishable from two-factor interactions, allowing for the separate estimation of quadratic terms without interference. More details can be found in Goos and Jones [20].

Definitive Screening Designs can be implemented directly by using conference matrices, see the work of Xiao et al. [15]. The *DSD* design, based on the application of Conference Matrices $C_m$, could be of the form

$$DSD = \begin{pmatrix} C_m \\ -C_m \\ 0_s \end{pmatrix}$$

with $C_m$ satisfying the equation $C'_m C_m = (m-1)I_m$.

**Example 5** An example of a Definitive Screening Design of order 4 is:

$$DSD_{9\times4} = \begin{pmatrix} C_4 \\ -C_4 \\ 0_1 \end{pmatrix} = \left( \begin{array}{cccc} 0 & 1 & 1 & 1 \\ -1 & 0 & -1 & 1 \\ -1 & 1 & 0 & -1 \\ -1 & -1 & 1 & 0 \\ \hline 0 & -1 & -1 & -1 \\ 1 & 0 & 1 & -1 \\ 1 & -1 & 0 & 1 \\ 1 & 1 & -1 & 0 \\ \hline 0 & 0 & 0 & 0 \end{array} \right)$$

where $C_4$ is the Conference matrix of order 4.

In their work, Nguyen and Stylianou [21] highlight that employing a Conference matrix $C_m$ of order $m$, which corresponds to a weighing matrix $W_m$ of the same order and weight $m-1$, results in a Definitive Screening Design (DSD) that maintains orthogonality among its main effects. They also present an algorithm that allows for the generation of Definitive Screening Designs (DSDs) even when the number of factors is odd. This ensures that DSDs can be created for both even and odd values of $m$. One significant recent paper, see Ares and Goos [22], that deals with the blocking of DSDs appear in the literature. The reader who is interested in more details in DSDs may also consider the paper Ares and Goos [22] and the references therein.

## 3.6 Orthogonal matrices in block structure $T_{OD}$

Using the approach outlined in the section on block structure of Orthogonal matrices by Alrweili et al. [9], we have generated new composed Orthogonal designs. Those designs consist of the blocks of known OD that are combined in such a way that they maintain their orthogonality. The following example illustrates this methodology and generated orthogonal designs that will be used as building blocks in composite designs.

**Example 6** Consider the familiar orthogonal designs $OD_2$, $OD_4$, and $OD_6$ as described in the Orthogonal Designs section. New block orthogonal designs of order 12 can be constructed by using a combination of those designs and filling the missing entries with zeros. These include the following three block orthogonal designs of order 12 that are of a completely

different structure.

$$\frac{T_{OD_{12a}}}{\begin{pmatrix} OD_2 & 0_{2\times4} & 0_{2\times6} \\ 0_{4\times2} & OD_4 & 0_{4\times6} \\ 0_{6\times2} & 0_{6\times2} & OD_6 \end{pmatrix}} \qquad \frac{T_{OD_{12b}}}{\begin{pmatrix} OD_4 & 0_{4\times4} & 0_{4\times4} \\ 0_{4\times4} & OD_4 & 0_{4\times4} \\ 0_{4\times4} & 0_{4\times4} & OD_4 \end{pmatrix}} \qquad \frac{T_{OD_{12c}}}{\begin{pmatrix} OD_6 & 0_{6\times6} \\ 0_{6\times6} & OD_6 \end{pmatrix}}$$

Note that the block orthogonal design $T_{OD_{12a}}$ has 8 distinct variables, $T_{OD_{12b}}$ has 12 distinct variables and $T_{OD_{12c}}$ has 4 distinct variables.

The same methodology can also be applied in cases where we have an odd number of factors, we can add an extra variable $a$ and fill the missing value with $0_s$.

**Example 7** Suppose we want to create a block orthogonal design with 7 factors, denoted as $T_{OD_7}$. We can achieve this by starting with a block orthogonal design of order 6, such as $OD_{6(4+2)}$, composed of $OD_4$ and $OD_2$. Next, we introduce a new variable, let's call it $a$, in the $(7, 7)$ entry, filling the positions $(i, 7)$ and $(7, i)$ with zeros for all $i = 1 \dots, 6$. The resulting block orthogonal design will be:

$$\frac{T_{OD_7}}{\begin{pmatrix} OD_2 & 0_{2\times4} & 0_{2\times1} \\ 0_{4\times2} & OD_4 & 0_{4\times1} \\ 0_{1\times2} & 0_{1\times4} & a \end{pmatrix}}$$

These block orthogonal designs will be referred to as matrices $T_{OD_k}$, with $k$ representing the number of rows. This nomenclature helps differentiate them from the traditional orthogonal designs outlined in section 3.1. These matrices can serve as the axial components in certain composite designs.

## 4 Construction methodology

In this section, we'll explore four distinct design methods resulting from the combination of well-defined factorial and axial components. These methods draw from established experimental design literature. The selection of both the factorial and axial parts involves different approaches, aiming to create a derived design with enhanced efficiency. The first two methods are grounded in the framework initially discussed by Georgiou et al. [2].

The variations in building composite designs stem from specific choices made for the factorial and axial components. For instance, in the first and third methods, which we'll discuss later, the axial part comprises known orthogonal designs, primarily outlined in Seberry and Yamada [23]. In contrast, the second and fourth methods incorporate axial parts consisting of block-structured orthogonal designs mentioned earlier. Below, we provide a more in-depth exploration of each of the four approaches and an illustration for each.

## 4.1 Method 1

This method was first introduced and used in the work of Georgiou et al. [2]. The design matrix is

$$X = \begin{pmatrix} F \\ A \\ -A \\ C \end{pmatrix}.$$

The Factorial part $F$ is a 2-level matrix, usually defined by a number of columns of a Hadamard matrix. The axial part $A$ is a 3-level design, established by substituting the variables in the classic orthogonal designs $ODs$ from section 3.1 with 0, 1, −1. These differ from the conventional axial part of a CCD, as explained in Montgomery [24]. The orthogonality property is maintained in the design matrix $X_m$. Some new designs with larger D-values are presented in coded form in Table 1. To demonstrate this methodology, consider the following example.

**Example 8** The design matrix $X$ is denoted by $X = D_{H16A8a}$, and it is composed by the factorial part $F$, which consists of four columns of a Hadamard of order 16, and the axial part $A$, which consists of four columns of the orthogonal design $OD$ of order 8. So, the design matrix $X$ will include all these components as $X = [F^T, A^T, -A^T]^T$. The design Matrix $X$ has $k = 4$ factors and $n_r = 32$ runs, hence it has the coded name $D_{H16A8a}$. Keep in mind that the experimenter has the flexibility to add any desired number of center points to this matrix.

$$F = H_{16 \times 4} = \qquad [A_{8\times4}^T, -A_{8\times4}^T]^T$$

$$
\begin{pmatrix}
1 & 1 & 1 & 1 \\
-1 & 1 & -1 & 1 \\
1 & -1 & -1 & 1 \\
-1 & -1 & 1 & 1 \\
1 & 1 & 1 & -1 \\
-1 & 1 & -1 & -1 \\
1 & -1 & -1 & -1 \\
-1 & -1 & 1 & -1 \\
1 & 1 & 1 & 1 \\
-1 & 1 & -1 & 1 \\
1 & -1 & -1 & 1 \\
-1 & -1 & 1 & 1 \\
1 & 1 & -1 & -1 \\
-1 & 1 & -1 & -1 \\
1 & -1 & -1 & -1 \\
-1 & -1 & 1 & -1
\end{pmatrix}
\begin{pmatrix}
1 & 1 & 1 & 0 \\
-1 & 1 & 0 & -1 \\
-1 & 0 & 1 & 1 \\
0 & 1 & -1 & 1 \\
-1 & -1 & 1 & -1 \\
-1 & 1 & -1 & -1 \\
-1 & -1 & -1 & 1 \\
-1 & 1 & 1 & 1 \\
-1 & -1 & -1 & 0 \\
1 & 1 & 0 & 1 \\
1 & 0 & 1 & 1 \\
0 & -1 & 1 & -1 \\
1 & 1 & -1 & 1 \\
1 & -1 & 1 & 1 \\
1 & 1 & 1 & -1 \\
1 & -1 & -1 & -1
\end{pmatrix}
$$

**Table 1. New designs X = [F$^T$; A; -A].**

| Name | k, n$_r$, n$_c$ | Factorial part (F) | Axial part (A) | D-value |
|---|---|---|---|---|
| $D_{H16A8}$ | 6, 32, 0 | 728,188,512,80,708,168,492,60,720,180, 504,72,704,164,488,56 | 377,393,281,125,58,166,21, 235 | 359 |
| $D_{H20A8}$ | 6, 36, 0 | 728,60,168,656,492,18,56,26,216,504,240, 564,236,702,650,666,548,2,188 | 404,384,37,203,67,193,105, 477 | 383 |
| $D_{H24A8}$ | 6, 40, 0 | 728,236,704,224,564,72,540,60,648,168, 672,180,488,8,512,20,405,591,45,231,543, 721,323,137 | 683,497,185,7,697,175, 463,157 | 386 |
| $D_{H32A12}$ | 8, 56, 0 | 5897,2141,4965,4833,1287,3825,4616,722, 1698,60,163,23 | 5962,2093,5065,3132,2503, 1155,3149,2639,1131,355, 824,2230 | 345 |

## 4.2 Method 2

Similarly to the first method, in this construction, we opt for the factorial part to consist of selected columns from a Hadamard matrix, following the description in Method 1. The resulting design matrix is given by:

$$X = \begin{pmatrix} F \\ T_{OD} \\ -T_{OD} \end{pmatrix}.$$

In contrast to Method 1, the key difference here is the utilization of $T_{OD}$ as the axial part of the design matrix. These structures follow a block orthogonal pattern, as explained in section 3.6. The design matrices produced through this method, leading to enhanced efficiency, are displayed in coded format in Table 2.

**Example 9** The design matrix $D_{H12A5(3+2)}$ consists of four columns from a Hadamard matrix of order 12 as the factorial part and four columns from a block orthogonal structure $T_{OD(3+2)}$ of order 5 as the axial part. The parameters of $T_{OD(3+2)}$ indicate that its block structure is formed by employing a block orthogonal matrix of order 3 and a classical orthogonal design of order 2.

$$F = H_{12\times 4} = \begin{pmatrix} 1 & 1 & 1 & 1 \\ -1 & 1 & -1 & 1 \\ -1 & -1 & 1 & -1 \\ 1 & -1 & -1 & 1 \\ -1 & 1 & -1 & -1 \\ -1 & -1 & 1 & -1 \\ -1 & -1 & -1 & 1 \\ 1 & -1 & -1 & -1 \\ 1 & 1 & -1 & -1 \\ 1 & 1 & 1 & -1 \\ -1 & 1 & 1 & 1 \\ 1 & -1 & 1 & 1 \end{pmatrix} \qquad [T^T_{OD(3+2)}, -T^T_{OD(3+2)}]^T \begin{pmatrix} 1 & 1 & 0 & 0 \\ -1 & 1 & 0 & 0 \\ 0 & 0 & 1 & 0 \\ 0 & 0 & 0 & 1 \\ 0 & 0 & 0 & -1 \\ -1 & -1 & 0 & 0 \\ 1 & -1 & 0 & 0 \\ 0 & 0 & -1 & 1 \\ 0 & 0 & 0 & -1 \\ 0 & 0 & 0 & 1 \end{pmatrix}$$

**Table 2. New designs X[F;T_OD;- T_OD].**

| Name | k, n_r, n_c | Factorial part (F) | Block ODs (T_OD) | D-value |
|---|---|---|---|---|
| $D_{H8T5}$ | 4,18,2 | 80,20,56,8,78,18,54,6 | 53,11,21,7,40[cp] | 343 |
| $D_{H16aT5a}$ | 5,26,0 | 242,78,182,18,222,62,162,2,234,74,186,26,218,54,170,6 | 160,34,64,22,122 | 457 |
| $D_{H16T7a}$ | 6,30,2 | 728,188,512,80,708,168,492,60,720,180,504,72,704,164,488,56 | 671,241,557,47,267,89,364[cp] | 335 |
| $D_{H40T7b}$ | 6,54,2 | 728,54,222,654,488,18,2,72,170,560,188,560,240,708,710,672,486,56,188,504,720,62,218,650,492,26,6, 80,162,564,180,564,236,704,702,668,494,60,180,512 | 481,103,193,67,365,361,364[cp] | 442 |
| $D_{H32T7a}$ | 7,46,0 | 2186,564,1538,240,2130,512,1482,188,2166,548,1518,224,2114,492,1466,168,2178,560,1530,236,2126, 504,1478,180,2162,540,1514,216,2106,488,1458,164 | 1457,251,621,537,659,183,223 | 417 |
| $D_{H40T7b}$ | 7,54,0 | 2186,564,1538,240,2130,512,1482,188,2166,548,1518,224,2114,492,1466,168,2178,560,1530,236,2126, 504,1478,180,2162,540,1514,216,2106,488,1458,164,1457,251,621,537,659,183,223,74 | 729,1935,1565,1649,1527,2003 1963 | 436 |
| $D_{H32T13a}$ | 8,64,0 | 6560,648,1946,5832,4400,54,188,542,1646,4940,1698,5100,2162,6480,6342,5888,4538,510,1536,4590, 6552,656,1950,5840,4392,62,180,546,1638,4932,1694,5096,2166,488,6338,5892,4542,506,1532,4598 | 5962,2093,5065,3132,2503,1155,3149, 2639,1131,355,824,2230 | 403 |
| $D_{H40T13a}$ | 8,64,0 | 6560,648,1952,5852,4400,62,168,546,1622,4932,1680,5096,2162,6504,6342,5904,4554,488,1512,4616, 6534,674,1962,5838,4374,72,182,560,1644,4922,1694,5082,2184,6482,6320,5894,4544,510,1538,4590 | 3710,1411,4719,4752,1260,3816,4856, 1450,3885,69,190,104,3280[cp] | 407 |

[cp]: indicates that some center points were generated in this part of the design.

The composite design matrix $X$ features $k = 4$ factors, $n_r = 22$ runs, and was created using a Hadamard matrix of order 12 and a block orthogonal matrix of order 5 with type (3 + 2). Consequently, it has been assigned the coded label $D_{H12A5(3+2)}$.

## 4.3 Method 3

In the third method, we propose a novel composition for a composite design, comprising a factorial part, a Definitive Screening Design (DSD) part, and an axial part. The added DSD is built from a conference matrix $C_m$ of order $m$ without central points. In essence, our design takes the form:

$$X = \begin{pmatrix} F \\ C_m \\ -C_m \\ A \\ -A \end{pmatrix}.$$

Though there may seem to be a considerable number of runs, this innovative design demonstrates situations where it achieves a notably higher D-value while keeping run sizes comparable to those documented in the literature. Furthermore, these designs allow for the direct implementation of large-scale experiments without the need for a computer search, ensuring high efficiency, especially in cost-effective experimental settings.

**Example 10** The design matrix $D_{H8A4DSD4}$ is a composite design created with four columns from a Hadamard matrix of order 8, four columns from a Definitive Screening Design (DSD) with 4 factors generated from a conference matrix $C_{m_4}$, and an axial part $A$ generated from four columns of the classic orthogonal design of order 4. The resulting design matrix is as follows:

$$
\begin{array}{c}
F = H_{8\times4} \\
\begin{pmatrix}
1 & 1 & 1 & 1 \\
-1 & 1 & -1 & 1 \\
1 & -1 & -1 & 1 \\
-1 & -1 & 1 & 1 \\
1 & 1 & 1 & -1 \\
-1 & 1 & -1 & -1 \\
1 & -1 & -1 & -1 \\
-1 & -1 & 1 & -1
\end{pmatrix}
\end{array}
\quad
\begin{array}{c}
C_m = Cm_{4\times4} \\
\begin{pmatrix}
0 & 1 & 1 & 1 \\
-1 & 0 & -1 & 1 \\
-1 & 1 & 0 & -1 \\
-1 & -1 & 1 & 0
\end{pmatrix}
\end{array}
\quad
\begin{array}{c}
A = A_{4\times4} \\
\begin{pmatrix}
1 & 1 & 0 & 1 \\
-1 & 1 & -1 & 0 \\
0 & 1 & 1 & -1 \\
-1 & 0 & 1 & 1
\end{pmatrix}
\end{array}
$$

The Table 3 below demonstrates such new design matrices in coded form.

## 4.4 Method 4

The fourth method involves a novel construction that combines aspects of methods two and three. This innovative structure introduces new design matrices with larger D-values than those found in the existing literature. The design comprises a Factorial part, a Definitive Screening Design (DSD) matrix, and an Axial part. Like the third method, this approach incorporates a DSD constructed from a conference matrix $C_m$ without central points. Additionally, the Axial part can be generated as a block orthogonal matrix, similar to the implementation in

**Table 3. New designs X[F;DSD;A; -A].**

| Name | k, $n_r$, $n_c$ | Factorial part (F) | Conf. Matrix ($C_m$) | Axial part (A) | D-value |
|------|------|------|------|------|------|
| $D_{H32Cm7A12}$ | 7, 72, 0 | 2186, 564, 1538, 240, 2130, 512, 1482, 188, 2166, 548, 1518, 224, 2114, 492, 1466, 168, 2178, 560, 1530, 236, 2126, 504, 1478, 180, 2162, 540, 1514, 216, 2106, 488, 1458, 164 | 1457, 251, 621, 537, 659, 183, 223, 74 | 1987, 697, 1688, 1044, 834, 385, 1049, 879, 377, 118, 274, 743 | 448 |
| $D_{H20Cm9A20}$ | 9, 80, 0 | 19682, 1968, 5858, 17574, 13140, 218, 494, 1692, 4914, 14802, 5100, 15248, 6480, 19496, 18954, 17666, 13682, 1520, 4556, 13776 | 13121, 15335, 14069, 13605, 15153, 15113, 18045, 18149, 19015, 19134 | 18974, 6379, 2127, 13835, 17691, 14739, 4805, 14345, 3863, 11493, 242, 564, 1532, 4436, 13148, 162, 488, 1464, 4392, 13176 | 329 |
| $D_{H32Cm9A20}$ | 9, 92, 2 | 19682, 4938, 13794, 1970, 18978, 4562, 13202, 1698, 19662, 4922, 13778, 950, 18962, 4542, 13182, 1682, 19674, 4934, 13790, 1962, 18974, 4554, 13194, 1694, 19658, 4914, 13770, 1946, 18954, 4538, 13178, 1674 | 13121, 15335, 14069, 13605, 15153, 15113, 18045, 18149, 19015, 19134 | 12413, 4192, 1398, 13592, 17610, 14712, 4796, 14342, 3862, 11493, 242, 564, 1532, 4436, 13148, 162, 488, 1464, 4392, 13176 | 395 |
| $D_{H32C10A20}$ | 10, 92, 0 | 59048, 14768, 41336, 5912, 56888, 13688, 39608, 5048, 59028, 14748, 41316, 5892, 56868, 13668, 39588, 5028, 59040, 14760, 41328, 5904, 56880, 13680, 39600, 5040, 59024, 14744, 41312, 5888, 56864, 13664, 39584, 5024 | 39365, 46007, 42209, 40815, 45461, 45339, 54137, 54447, 57045, 57403 | 3724, 12576, 4196, 40776, 52832, 44136, 14388, 43028, 11588, 34480, 726, 1694, 4598, 13310, 39446, 488, 1464, 4392, 13176, 39528 | 326 |

the third method. The orthogonality property of the factors is preserved in this method as well. The structure of the suggested design is given below:

$$X = \begin{pmatrix} F \\ C_m \\ -C_m \\ T_{OD} \\ -T_{OD} \end{pmatrix}.$$

**Example 11** The design matrix, denoted as $D_{H16A8(6b+2)DSD4}$, is constructed by combining three components. These components include four columns from a Hadamard matrix of order 16, four columns from a definitive screening design created using a conference matrix $Cm_4$, and four columns from a block orthogonal matrix $T_{OD} = T_{OD(6+2)}$ of order 8. The parameters in $T_{OD(6+2)}$ signify the use of orthogonal designs of order 6 and 2 in forming the block structure, as detailed in section 3.6.

$$F = H_{16 \times 4}$$
$$\begin{pmatrix} 1 & 1 & 1 & 1 \\ -1 & 1 & -1 & 1 \\ 1 & -1 & -1 & 1 \\ -1 & -1 & 1 & 1 \\ 1 & 1 & 1 & -1 \\ -1 & 1 & -1 & -1 \\ 1 & -1 & -1 & -1 \\ -1 & -1 & 1 & -1 \\ 1 & 1 & 1 & 1 \\ -1 & 1 & -1 & 1 \\ 1 & -1 & -1 & 1 \\ -1 & -1 & 1 & 1 \\ 1 & 1 & 1 & -1 \\ -1 & 1 & -1 & -1 \\ 1 & -1 & -1 & -1 \\ -1 & -1 & 1 & -1 \end{pmatrix}$$

$$C_m = Cm_{4 \times 4}$$
$$\begin{pmatrix} 0 & 1 & 1 & 1 \\ -1 & 0 & -1 & 1 \\ -1 & 1 & 0 & -1 \\ -1 & -1 & 1 & 0 \end{pmatrix}$$

$$T_{OD} = T_{OD(6+2)}$$
$$\begin{pmatrix} 1 & 0 & 1 & -1 \\ 1 & 1 & 0 & 1 \\ 0 & 1 & 1 & 0 \\ 1 & -1 & 0 & 1 \\ 0 & 1 & -1 & 0 \\ -1 & 0 & 1 & 1 \\ 0 & 0 & 0 & 0 \\ 0 & 0 & 0 & 0 \end{pmatrix}$$

**Table 4. New designs X[F;DSD;T_OD;- T_OD].**

| Name | k, $n_r$, $n_c$ | Factorial part(F) | Conf.Matrix ($C_m$) | Block ODs ($T_{OD}$) | D-value |
|---|---|---|---|---|---|
| $D_{H16aCm5T5}$ | 5,38,0 | 242,78,182,18,222,62,162,2,234,74,186, 26,162,2,234,74,186,26,218,54,170,6 | 54,170,6,74,180,240 | 78,6,54,18,2 | 458 |
| $D_{H32Cm6T7}$ | 6,58,2 | 728,236,560,80,668,188,512,20,704,224, 548,56,656,164,488,8,726,234,558,78,666, 186,510,18,702,222,546,54,654,162,486,6 | 405,591,45,231,543,721 | 671,241,557,47,267,89,364$^{cp}$ | 462 |
| $D_{H20Cm7T7}$ | 7,50,0 | 2186,564,1538,240,2130,512,1482,188,2166, 548,1518,224,2114,492,1466,168,2178,560, 1530,236,2126,504,1478,180,2162,540,1514, 216,2106,488,1458,164 | 1457,251,621,537,659,183,223,74 | 2014,724,1672,142,802,268,1094 | 378 |
| $D_{H32Cm7T7}$ | 7,62,2 | 2186,564,1538,240,2130,512,1482,188,2166, 548,1518,224,2114,492,1466,168,2178,560, 1530,236,2126,504,1478,180,2162,540,1514, 216,2106,488,1458,164,1457,251,621,537,659, 183,223,74 | 6,504,1478,180,2162,540,1514, 216,2106,488,1458,164 | 2014,724,1672,142,802,268,1094 | 445 |
| $D_{H20Cm9T21}$ | 9,82,2 | 19682,1952,5910,17682,13202,234,540,1682, 5042,14742,5082,15090,6336,19658,19008, 17498,13776,1640,4382,13682 | 13121,15335,14069,13605,15153, 15113,18045,18149,19015,19134 | 12413,4193,1401,13601,17637,14631, 4553,13613,1675,4932,242,564,1532, 44361,3148,162,488,1464,4392,13176,9841$^{cp}$ | 339 |
| $D_{H32Cm9T18}$ | 9,88,4 | 19682,4938,13794,1970,18978,4562,13202, 1698,19662,4922,13778,950,18962,4542, 13182,1682,19674,4934,13790,1962,18974, 4554,13194,1694,19658,4914,13770,1946, 18954,4538,13178,1674 | 13121,15335,14069,13605,15153, 15113,18045,18149,19015,19134 | 19518,5084,6020,2180,13363,1518, 4446,510,13283,1677,4551,693, 6563,1727,5177,2735,9841$^{cp}$,9841$^{cp}$ | 371 |
| $D_{H32Cm9T19}$ | 9,90,6 | 19682,4938,13794,1970,18978,4562,13202, 1698,19662,4922,13778,950,18962,4542,13182, 1682,19674,4934,13790,1962,18974,4554,13194, 1694,19658,4914,13770,1946,18954,4538, 13178,1674 | 13121,15335,14069,13605,15153, 15113,18045,18149,19015,19134 | 19519,5083,6019,2179,13363,1518, 4446,510,9881,9939,9897,9927,6683, 1823,5225,2795,9841$^{cp}$,9841$^{cp}$,9841$^{cp}$ | 375 |
| $D_{H32Cm9T21}$ | 9,94,2 | 19682,4938,13794,1970,18978,4562,13202,1698, 19662,4922,13778,950,18962,4542,13182,1682, 19674,4934,13790,1962,18974,4554,13194,1694, 19658,4914,13770,1946,18954,4538,13178,1674 | 13121,15335,14069,13605,15153, 15113,18045,18149,19015,19134 | 18974,6379,2127,13835,17691,14739, 4805,14345,3863,11493,242,564,1532,4436, 13148,162,488,1464,4392,13176,9841$^{cp}$ | 399 |
| $D_{H32Cm10T21}$ | 10,94,2 | 59048,14768,41336,5912,56888,13688,39608,5048, 59028,14748,41316,5892,56868,13668,39588,5028, 59040,14760,41328,5904,56880,13680,39600,5040, 59024,14744,41312,5888,56864,13664,39584,5024 | 39365,46007,42209,40815,45461, 45339,54137,54447,57045,57403 | 37241,12579,4205,40803,52913,43893, 13659,40841,5027,14797,726,1694,4598,13310, 39446,488,1464,4392,13176,39528,29524$^{cp}$ | 330 |

$^{cp}$: indicates that some center points were generated in this part of the design.

The label for the final design matrix $X$ in this example, which incorporates $k = 4$ factors and $n_r = 40$ runs, is designated as $D_{H16A8(6+2)DSD4}$. This label reflects the components employed in constructing the matrix, including a Hadamard matrix of order 16, a definitive screening design of 4 factors, and a block orthogonal matrix of order 8 and type (6 + 2).

Table 4 displays the design matrices in coded form. Utilizing the recommended methods, we generate alternative composite designs suitable for response surface methodology. Details on these constructed designs and their associated D-values are presented in the following section.

## 5 Composite designs comparison

In this section, we display the outcomes in Tables 1–4. The initial column in these tables signifies the name (label) of the design. This name comprises parameters describing the components of the specific design matrix. Specifically, the design matrix named $D_{H_nA_kDSD_l}$ denotes construction using some columns of the Hadamard matrix of order $n$, an equal number of columns from an orthogonal design of order $k$ as its axial part, and the same number of columns from a Definitive Screening Design of order $l$.

The parameters in the subsequent three columns of the tables $k$, $n_r$, and $n_c$ represent the number of factors, the number of runs (excluding center points), and the number of center points in the design matrix, respectively. The succeeding columns present the factorial ($F$), axial ($A$), and/or block Orthogonal Design ($T_{OD}$) in coded form. In Table 3, the column *Conf. Matrix*($C_m$) presents the conference matrix in coded form, crucial for constructing the required Definitive Screening Design.

The coded form of a design is derived by replacing 1$s$ with 2$s$, 0$s$ with 1$s$, and $-1s$ with 0$s$, creating a number in the Ternary numeric system for each run of the design. Subsequently, the ternary value for each run is converted into the decimal numeric system, with missing digits replaced by 0s in the ternary representation.

The concluding columns of the tables furnish the calculated $D$-values of the model matrix for each design using the formula (3), as provided by Nguyen and Lin [10].

**Example 12** The deconstruction method is elucidated below. Suppose we aim to reconstruct the $D_{H16Cm5T5}$ design, as illustrated in Table 4. This design comprises the factorial part ($F$), the conference matrix ($C_m$), and the block orthogonal design matrix ($T_{OD}$). It is a five-factor design ($k = 5$) with a total of 38 runs ($n_r = 38$) and no center points ($n_c = 0$). Each decimal number corresponds to a specific run in the design matrix. To decode, simply convert the decimal numbers into ternary numbers; for example, $170_{10} = 20022_3$. Then, subtract one from each ternary digit to obtain the corresponding run in the design matrix. In this instance, $20022_3$ results in the design's run $+ 1, -1, -1, + 1, + 1$.

For conference matrices, other axial points, or block orthogonal designs, the notation $-C_m$, $-A$, or $-T_{OD}$ denotes the negative of the matrix that is constructed by converting all $+ 1s$ to $-1s$ while leaving the 0$s$ unchanged.

## 6 Discussion

Creating composite designs is generally considered a manageable task, but the selection of columns in the design matrix becomes more intricate, especially when dealing with a larger number of factors and runs. The strategy proposed in this study not only proves to be easy to implement but also highly adaptable, allowing for the generation of designs tailored to the specific requirements of the experiment. A critical evaluation criterion for composite designs is the D-value, where higher values signify a more reliable and efficient model.

In the context of implementing 2nd order designs while maintaining orthogonality among main effects, four distinct methods were introduced. The first two methods, drawn from existing literature, have been applied in this paper, leading to the discovery of new designs that exhibit higher D-values than those reported in the existing literature.

Methods 3 and 4 introduce two entirely new techniques in experimental design. Although these approaches may necessitate more runs in some cases, they consistently yield much higher D-values than those found in the literature. An effort is consistently made to minimize the number of runs while ensuring the effectiveness of the design.

Tables 1–4 provide detailed insights into all the designs developed in this study. Further comparisons are presented in Table 5, where these designs are juxtaposed with existing composite designs from the literature. The table includes the authors' names, the parameter $k$ denoting the number of factors, and rounded D-values, along with the total run sizes of the composite designs in parentheses.

In the last row of Table 5, our newly proposed designs are highlighted, showcasing either significantly higher D-values or a reduced number of runs compared to the designs listed above. This suggests that the results presented in this paper have the potential to effectively reduce the cost of experiments while still achieving desired outcomes within a satisfactory time

**Table 5. Comparison of D-values.**

| Literature | k: 4 | 5 | 6 | 7 | 8 | 9 | 10 |
|---|---|---|---|---|---|---|---|
| Box and Wilson (1951) | 457(24) | 440(26) | 456(44) | 465(78) | 474(80) | 480(146) | 493(148) |
| Draper and Lin (1990) | 308(16) | 24(21) | NE | 197(36) | 221(46) | 200(56) | 165(66) |
| Morris(2000) | 373(36) | 308(26) | 298(36) | 269(36) | 272(78) | 252(78) | 238(78) |
| Angelopoulos et al (2009) | 308(16) | 178(21), 256(22) | NE | 234(36) | 242(46) | 222(56) | NE |
| Nguyen and Lin (2011) | NE | 259(22), 355(26) | 263(28), 368(36) | 262(38) | 280(48), 252 (48) | 246(48) | 224(68) |
| Georgiou et al (2014) | 339(16), 395(20) | 181(21), 276(22), 322 (24), 440(26), 425 (36) | 331(28), 379(36) | 291(36), 348(38) | 129(45), 317 (46), 339(48) | 15(55), 313(56), 314(58) | 302(66), 310(68) |
| Alrweili et al (2020) | NE | 311(22), 457(26), 424 (30), 448(34) | NE | 348(38) | NE | 153(55) | NE |
| *New designs* | 343(18)$^\S$, 459(44)$^\flat$ | 458(38)$^\ddagger$ | 335(30)$^\S$, 359(32)$^\flat$, 383 (36)$^\flat$, 407(40)$^\flat$, 442(54)$^\S$, 462(58)$^\ddagger$ | 416(46)$^\S$, 436(54)$^\S$, 445(62)$^\ddagger$, 449(72)$^\sharp$ | 345(56)$^\flat$, 403 (64)$^\flat$, 407(66)$^\S$ | 329(80)$^\sharp$, 339(82)‡, 372 (88)$^\ddagger$, 375(90)$^\ddagger$, 395(92)$^\sharp$, 399(94)$^\ddagger$ | 326(92)$^\sharp$, 330(94)$^\ddagger$ |

Each symbol indicates the implementation method for each design.

$^\flat$: method 1,

$^{\#x00A7}$: method 2,

$^\sharp$: method 3, and

$^\ddagger$: method 4

frame. All four methods are capable of producing designs with the same (or similar) number of runs as existing composite designs but are more efficient in terms of D-value, using the same design parameters outlined in the literature.

The ideas and methods shared in this paper offer possibilities for future research in experimental design. One important area is testing the proposed methods with more factors, going beyond the current limit of eight. Exploring how well these methods work with a higher number of factors will help us understand their adaptability and effectiveness in more complex experimental situations. Additionally, studying designs with more than eight factors in greater detail can give us more insights into the practical benefits and limits of the proposed approaches.

Another interesting direction is to continue improving and adjusting the methods based on real-world applications and feedback. Ongoing research could focus on using advanced statistical and computational techniques to refine design matrices, possibly reducing the number of runs while improving overall performance. This ongoing process of improvement, along with further exploration of the methods, will contribute to advancing and making experimental design methods more practical.

Building on what we have learned in this paper, there are more ways to explore and improve experimental design methods. We could try different versions of the proposed methods, maybe using different statistical models or adjusting them for specific experiments. Testing how well these methods work in various types of experiments and different conditions can help us understand their usefulness more broadly. Also, comparing them with other design strategies or getting feedback from people using them in real situations can give us more insights. These additional research paths aim to make our understanding of experimental design better and keep making improvements.

## Acknowledgments

The authors would like to thank the two anonymous referees and the associate editor for their valuable comments and suggestions that improved the results and the appearance of the paper.

## Author Contributions

**Methodology:** Despina E. Athanasaki, Stelios D. Georgiou, Stella Stylianou.

**Software:** Despina E. Athanasaki, Stelios D. Georgiou.

**Supervision:** Stelios D. Georgiou, Stella Stylianou.

**Writing – original draft:** Despina E. Athanasaki.

**Writing – review & editing:** Stelios D. Georgiou, Stella Stylianou.

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
