## [Decision Letter · Decision Letter 0]

16 Jan 2024

PONE-D-23-40932New approaches on Composite Designs for Response Surface MethodologyPLOS ONE

Dear Dr. Georgiou,

Thank you for submitting your manuscript to PLOS ONE. After careful consideration, we feel that it has merit but does not fully meet PLOS ONE’s publication criteria as it currently stands. Therefore, we invite you to submit a revised version of the manuscript that addresses the points raised during the review process.

We look forward to receiving your revised manuscript.

Kind regards,

Bohui Wang

Academic Editor

PLOS ONE

Journal Requirements:

Additional Editor Comments:

This paper needs a major revision to improve the presentation.

Reviewers' comments:

Reviewer's Responses to Questions

**Comments to the Author**

1. Is the manuscript technically sound, and do the data support the conclusions?

Reviewer #1: Yes

Reviewer #2: Yes

2. Has the statistical analysis been performed appropriately and rigorously? 

Reviewer #1: Yes

Reviewer #2: Yes

3. Have the authors made all data underlying the findings in their manuscript fully available?

Reviewer #1: Yes

Reviewer #2: Yes

4. Is the manuscript presented in an intelligible fashion and written in standard English?

Reviewer #1: Yes

Reviewer #2: Yes

5. Review Comments to the Author

Reviewer #1: "New Approaches on Composite Designs for Response Surface Methodology"

Major Revision required

I have carefully reviewed the above paper titled "New Approaches on Composite Designs for Response Surface Methodology." The authors introduce two novel composite techniques aiming to enhance design efficiency within Response Surface Methodology (RSM), with a particular emphasis on the importance of Definitive Screening Designs (DSDs) as an alternative to traditional axial points in composite designs.

The construction of new design matrices involves the use of orthogonal matrices and the integration of axial components from simple orthogonal designs or block orthogonal designs, as outlined by Alrweili in 2020. Significantly, the proposed designs exhibit superior performance in terms of efficiency and robustness compared to existing designs. The paper navigates the challenges associated with creating composite designs, especially when dealing with a larger number of factors and runs.

The implementation of second-order designs, while maintaining orthogonality among main effects, introduces four distinct methods. The application of the first two methods from existing literature leads to the discovery of new designs with higher D-values, while Methods 3 and 4 introduce entirely new techniques, consistently yielding higher D-values with a simultaneous reduction in the number of runs.

Detailed insights into all the designs are presented in tables, facilitating a thorough comparison with existing composite designs from the literature. Notably, the newly proposed designs are highlighted in the last row of the comparison table, showcasing higher D-values or a reduced number of runs. This suggests the potential for cost reduction in experiments while achieving desired outcomes within an acceptable time frame.

The paper not only makes a significant contribution to the field of experimental design within Response Surface Methodology but also offers practical benefits to readers and practitioners. The innovative approaches, coupled with a comprehensive comparative analysis, provide valuable insights for researchers and professionals engaged in experimental design. The potential for cost reduction highlighted in the findings could be particularly advantageous to practitioners in various fields.

The innovative techniques that are developed in the paper and their demonstrated advantages make it a valuable addition to the existing literature, providing a promising avenue for further research and practical applications in experimental design.

In conclusion, considering the substantial contribution of this paper to the field of designs of experiments and its potential benefits to readers and practitioners, I strongly recommend the acceptance of this paper for publication.

Reviewer #2: Manuscript is well written and accepted in present form and accepted for publication only reference are outdated. Kindly add updated references. Also mentioned how we validate the new developed tool as previously RSM updated through p and R values.

6. PLOS authors have the option to publish the peer review history of their article (what does this mean?). If published, this will include your full peer review and any attached files.

Reviewer #1: No

Reviewer #2: No

---

## [Author Response · Author response to Decision Letter 0]

21 Feb 2024

All the comments have been answered in the attached letter.

---

## [Decision Letter · Decision Letter 1]

11 Mar 2024

New approaches on Composite Designs for Response Surface Methodology

PONE-D-23-40932R1

Dear Dr. Georgiou,

We’re pleased to inform you that your manuscript has been judged scientifically suitable for publication and will be formally accepted for publication once it meets all outstanding technical requirements.

Kind regards,

Bohui Wang

Academic Editor

PLOS ONE

Additional Editor Comments (optional):

This paper can be acceptable now

Reviewers' comments:

Reviewer's Responses to Questions

**Comments to the Author**

1. If the authors have adequately addressed your comments raised in a previous round of review and you feel that this manuscript is now acceptable for publication, you may indicate that here to bypass the “Comments to the Author” section, enter your conflict of interest statement in the “Confidential to Editor” section, and submit your "Accept" recommendation.

Reviewer #1: All comments have been addressed

Reviewer #2: All comments have been addressed

2. Is the manuscript technically sound, and do the data support the conclusions?

Reviewer #1: Yes

Reviewer #2: Yes

3. Has the statistical analysis been performed appropriately and rigorously? 

Reviewer #1: Yes

Reviewer #2: Yes

4. Have the authors made all data underlying the findings in their manuscript fully available?

Reviewer #1: Yes

Reviewer #2: Yes

5. Is the manuscript presented in an intelligible fashion and written in standard English?

Reviewer #1: Yes

Reviewer #2: Yes

6. Review Comments to the Author

Reviewer #1: I have carefully reviewed the above paper titled "New Approaches on Composite Designs for Response Surface Methodology." The authors introduce two novel composite techniques aiming to enhance design efficiency within Response Surface Methodology (RSM), with a particular emphasis on the importance of Definitive Screening Designs (DSDs) as an alternative to traditional axial points in composite designs.

The construction of new design matrices involves the use of orthogonal matrices and the integration of axial components from simple orthogonal designs or block orthogonal designs, as outlined by Alrweili in 2020. Significantly, the proposed designs exhibit superior performance in terms of efficiency and robustness compared to existing designs. The paper navigates the challenges associated with creating composite designs, especially when dealing with a larger number of factors and runs.

The implementation of second-order designs, while maintaining orthogonality among main effects, introduces four distinct methods. The application of the first two methods from existing literature leads to the discovery of new designs with higher D-values, while Methods 3 and 4 introduce entirely new techniques, consistently yielding higher D-values with a simultaneous reduction in the number of runs.

Detailed insights into all the designs are presented in tables, facilitating a thorough comparison with existing composite designs from the literature. Notably, the newly proposed designs are highlighted in the last row of the comparison table, showcasing higher D-values or a reduced number of runs. This suggests the potential for cost reduction in experiments while achieving desired outcomes within an acceptable time frame.

The paper not only makes a significant contribution to the field of experimental design within Response Surface Methodology but also offers practical benefits to readers and practitioners. The innovative approaches, coupled with a comprehensive comparative analysis, provide valuable insights for researchers and professionals engaged in experimental design. The potential for cost reduction highlighted in the findings could be particularly advantageous to practitioners in various fields.

The innovative techniques that are developed in the paper and their demonstrated advantages make it a valuable addition to the existing literature, providing a promising avenue for further research and practical applications in experimental design.

In conclusion, considering the substantial contribution of this paper to the field of designs of experiments and its potential benefits to readers and practitioners, I strongly recommend the acceptance of this paper for publication.

Reviewer #2: The authors has incorporated all necessary changes which is suggested for improvement. Now the manuscript is ready for publication.

7. PLOS authors have the option to publish the peer review history of their article (what does this mean?). If published, this will include your full peer review and any attached files.

Reviewer #1: No

Reviewer #2: No

---

## [Editor Report · Acceptance letter]

21 Mar 2024

PONE-D-23-40932R1 

PLOS ONE

Dear Dr. Georgiou, 

I'm pleased to inform you that your manuscript has been deemed suitable for publication in PLOS ONE. Congratulations! Your manuscript is now being handed over to our production team.

Kind regards, 

on behalf of

Professor Bohui Wang 

Academic Editor

PLOS ONE